# An investigation of the diachronic trend of dependency distance minimization in magazines and news

**Ruoyang Zhang, Guijun Zhou** *

School of Foreign Languages, Northeast Normal University, Changchun, Jilin, PRC

* zhangry954@nenu.edu.cn

## Abstract

The principle of minimization of dependency distance (DD) can reduce the working memory burden of language speakers, that is, reduce the cognitive burden during the communication process. This investigation demonstrated the dependency distance (based on the dependency grammar) minimization principle from a diachronic perspective in two text types of magazines and news with inspection indicators of mean dependency distance (MDD) and normalized dependency distance (NDD). This research revealed a fluctuation tendency around a certain axis concerning diachronic dependency distance variation. This research also indicated that news text balances language complexity and communication efficiency better than magazine text.

## Introduction

American linguist George Kingsley Zipf, based on years of research on human behavior, proposed that human behavior follows the Principle of Least Effort, known as Zipf's Law. The principle of saving effort is reflected in the behavior of individuals and members of social groups, e.g., in language. Zipf explored the relationship between the frequency of different words in *Ulysses* and pointed out that the frequency ordering of a particular word multiplied by its frequency was a constant [1]. The words with the highest frequency are mostly short-syllable words, and the low-frequency ones are mostly long and multi-syllable words. People prefer short words for efficient communication. Through the high-frequency use of short syllable words, humans have achieved the effect of saving effort. Recently, studies have demonstrated the widespread presence of the effort-saving principle in language in various dimensions. One of the previous research directions is to demonstrate the dependency distance (based on the dependency grammar) minimization principle in human language use. Related research supports the hypothesis that the minimization of dependency distance is a universal phenomenon of natural languages [2].

Dependency grammar was proposed by French linguist Lucien Tesnière. As a French language teacher [3], he created a strict methodological framework, put forward the concept of dependency grammar, and applied it widely in the process of language teaching [4]. The influence of this grammar is currently widely spread owing to the modern advances in computer

**Data Availability Statement:** All relevant data are within the paper and its Supporting Information files.

**Funding:** The authors received no specific funding for this work.

**Competing interests:** No authors have competing interests. This does not alter our adherence to PLOS ONE policies on sharing data and materials.

**Fig 1. Dependency parsing tree.**

technology. The syntactic structure under this grammar enables the computer to automatically establish an analysis model. The author gives an example to illustrate the interdisciplinary combination of dependency grammar and computer application. Taking the automatic dependency analysis of the sentence, "*The quick brown fox jumped over the lazy dog*" as an example (Fig 1), the dependency grammar follows the syntax analysis [5].

1. In a sentence, there is only one root node, and the root node is generally the center verb (e.g., *jumped*) of the sentence. After we performed dependency parsing and annotation on the sentence, we obtained a dependency tree, as shown in the graph. A corpus annotated with dependency grammar is called a dependency treebank.

2. Except for the root node, all other words in the sentence have a parent node (for example, the parent node of *the* is *fox*, and the parent node of *fox* is *jumped*) and depend on this parent node.

3. There is a unique path from the parent node to the child node. More precisely, a word in the sentence has only one parent node.

In this example, *fox* is the nominal subject of *jumped* (nsubj: nominal subject). In *The quick brown*, *the* is the determiner of the noun subject *fox* (det: determiner), and both *quick* and *brown* are the modifiers of the noun subject *fox* (nmod: nominal modifier). In the second half of the sentence, the period is a direct dependent on the root node *jumped* of the sentence, *dog* is a nominal adjunct (obl: oblique nominal) of the root node *jumped*, and the relationship between *dog* and *jumped* is marked by *over* (case: case marking) as further explanation. Considering *the lazy dog*, *the* is the determiner of *dog* (det: determiner), and *lazy* is the adjective modifier of *dog* (amod: adjectival modifier). The entire sentence takes *jumped* as the root node and continues to branch out. The parent node controls the child node, and the child node depends on the parent node (governor) as a dependent; thus, it is called dependency grammar. However, the child node and the parent node are not always adjacent. For example, between the root node *jumped* and *dog*, if the linear position of *jumped* is 1 and the position of *dog* is 5, then the mutual distance is 4, which is called the dependency distance.

By calculating the average of the dependency distances of each dependency pair in the sentence, this method can accordingly calculate the average of the dependency distances of all dependency pairs of a specific sentence, text, or even the entire corpus. The mean dependency distance (MDD) is an important criterion for measuring the complexity of the syntax. The smaller the MDD, the more concise the sentence, and vice versa. The principle of minimization of dependency distance can reduce the working memory burden of language speakers, that is, reduce the cognitive burden on both sides of the communication process. The dependency distance and the assumption upon its minimization have been discussed widely in the academic field.

One theme is to examine the mechanism of dependency distance minimization and its impact. Some studies compared the dependency distance of the human natural language with

the dependency distance of languages randomly generated according to specific rules. Studies have revealed that the human natural language tends to reduce the dependency distance due to the reduction of the burden of working memory [6, 7]. Another research discusses how the change of dependency structure can better minimize the dependency distance and proposes an idealized method for the minimization of dependency distance [8]. Some relevant study is to examine the diachronic change in the average dependency distance of a specific language and determine what type of diachronic change it has experienced. Scholars used mean dependency distance (MDD) and normalized dependency distance (NDD) as indicators and based on the study of the diachronic dependence distance change of the State of the Union address in the United States, it was concluded that the dependence distance experience decreased over a long period [9]. This study, as the initial study on the diachronic change of the dependency distance, only examined the text type of the State of the Union addresses and the amount of data was relatively small. A subsequent study on other text types revealed that the dependency distance of different text types increased over time [10]. However, only MDD was used as an indicator, and the scope of the investigation was limited to sentences within a length of 10–30 words, which is not comprehensive enough to form a general idea of a certain tendency.

Another theme is to examine the factors that may affect dependency distance. Some studies examined the dependency distance between different genres or variants of texts in a specific language. For example, we may expect the dependency distance of spoken language to be smaller than that of the language of a novel written in English. The study concluded that genre had a minor effect on dependency distance, indicating that dependency distance is mainly determined by generalized cognitive factors rather than genre factors [11]. Research in this area has also compared the distribution of dependency distances between different variants of a particular language and proved that different English variants exhibit considerable similarity, whereas their dependencies are hardly affected by variants [12]. Some scholars have examined the gap between the dependency distances in the monolingual corpus and the hybrid corpus and have concluded that the hybrid dependency distance is longer than the monolingual dependency distance [13]. It has been proven that the variation of the dependency distance of bilingual data is larger than that of monolingual data [14]. Some research is to examine the impact of sentences of different lengths on the dependency distance. It was concluded that the distribution of dependency distances for sentences of the same length differed from that of sentences with different lengths [15]. Furthermore, in short sentence sequences and phrases, because of the shortening of sentence length and the reduction of the cognitive cost associated with it, the pressure of minimizing the dependency distance was relieved, and the phenomenon of the dependency distance inversely increasing emerged [16].

The third theme is to discuss the relationship between the principle of minimization of dependency distance, working memory, and cognition [17]. In particular, some scholars have discussed the relationship between the working memory burden and cognition process by comparing consecutive interpreters and simultaneous interpreters during the translation process [18, 19]. It has been found that contrary to intuitive feeling, the dependency distance of the text produced by simultaneous interpretation is greater than that of consecutive interpretation texts, indicating that consecutive interpretation has higher cognition requirements than simultaneous interpretation [20]. In addition, research in this area has experimentally verified the judgment that people's central memory storage is limited to three to five meaningful units [21], which strongly supports assumptions that there is a threshold of less than three words in MDD and grammar plays an important role in constraining dependency distance.

The present study explored the diachronic changes of dependency distance in different text types of magazines and news, as well as with larger datasets and expanded the inspection scope without limiting sentence length, and accordingly added the NDD inspection indicator, which

is different from the MDD indicator because NDD takes into consideration the sentence length factor and the root word location factor in addition to the MDD [22].

## Research questions

This study addressed two research questions.

1. What is the overall diachronic trend of MDD and NDD in magazines and news texts?

2. What are the similarities and differences between the MDD and NDD trends between magazines and news texts and how to explain these similarities and differences?

# Research methods

## Materials

This research was based on the Corpus of Historical American English (COHA). The time frame covered by COHA varies by text types, with the overall coverage spanning from 1810 to 2009. More specifically, the corpus from 1990 and later belongs to the scope of another corpus called Corpus of Contemporary American English (COCA). This study focused on the examination of two text types, news (1860–2009) and magazines (1815–2009), and their corpus feature metrics are as follows (Table 1). From the statistical characteristics of the two corpuses, there is no significant difference between the two text types.

COHA was chosen because it is the largest genre-specific diachronic corpus publicly accessible. The classification of the subject matter is clear and the time span is large. Publicly available features can facilitate repeated verification by other scholars. The reason for choosing magazines and news text types is that these two text types have relatively uniform stylistic formats and writing specifications, in contrast to other fictional texts, which include diverse genres such as poetry, novels, and dramas. These text subtypes contain significantly different textual characteristics, such as multiple dialogues in dramas, resulting in short sentences. In the case of poetry, the accuracy of the automatic dependency parsing is low, which accordingly, does not meet the intention of this research to limit the variables of the data. Non-fictional texts, in contrast to fictional texts, also have the characteristics of various subtypes of texts. Therefore, this study focused on the types of magazines and news texts in the COHA corpus.

## Steps

The author first preprocessed the COHA corpus, and the preprocessing part included removing the sentences containing hidden symbols in the corpus. Hidden symbols refer to a small

**Table 1. Magazines and news corpus feature metrics.**

| Corpus feature metrics | Magazines texts | News texts |
|---|---|---|
| Number of sentences | 4,068,253 | 1,823,549 |
| Number of tokens | 80,140,323 | 33,430,102 |
| Average word length | 4.56 | 4.63 |
| Number of word types | 605,520 | 400,347 |
| Standard type-token ratio | 0.49 | 0.49 |
| Average sentence length | 19.7 | 18.33 |
| Average number of punctuations in one sentence | 3.43 | 3.24 |
| Average clause length | 5.74 | 5.66 |

subset of words in the corpus that are hidden in a specific pattern during the corpus's release to the public. Using this pattern, the publisher of the corpus can mark the digital fingerprint of the corpus user in the case of unauthorized leaks. For sentences with hidden symbols, because the integrity of the sentence is destroyed, it is impossible to perform accurate dependency syntax analysis on it. The author first used a program written in the Python language to eliminate these sentences according to regular expressions. After culling, the corpus still maintained more than 95% integrity. Next, the author used NLTK [23] to segment the entire corpus because each line in the original state of the corpus is a separate article. After sentence segmentation, the author obtained a corpus in which each line became an independent sentence to facilitate automatic dependency parsing in sentence units in the later stage.

After completing the preprocessing of the corpus, the author used a program written in the Python language to call the Standard CoreNLP 4.4.0 Client [5] to perform automatic dependency syntax analysis on the corpus text. This automated analysis is not 100% correct. However, manual analysis is difficult to perform on huge data, and ultimately, it is not 100% correct either. The value of machine learning, which is currently in full swing, lies in its ability to learn patterns based on existing small-scale data to be applied to large-scale data sets and expand the boundaries of human cognition and exploration. Scholars have demonstrated that the parsing process of the Stanford parser is reliable, and possible errors do not significantly affect the results [9]. Furthermore, considering the huge volume of the COHA corpus, the author randomly selected 30% of the original corpus sentences for analysis and generated an annotated corpus for data analysis based on the randomly sampled text (Table 2). Subsequently, the author examined two indicators of the annotated corpus, namely the average dependency distance (MDD) [6] and the normalized average dependency distance (NDD) [22].

The formula for calculating MDD is:

$$MDD = \frac{\sum_{i=1}^{n} DD_i}{n}$$

With the above example, "*The quick brown fox jumped over the lazy dog.*" punctuation marks (used to break sentences rather than to present meaning itself) and the root word itself (which does not depend on any component and thus, there is no dependency distance or the dependency distance is 0) is out of the equation of dependency distance calculation.

Thus, the mean dependency distance (MDD) of the example is (Table 3):

(3 + 2 + 1 + 1 + 3 + 2 + 1 + 4)/8 = 2.125

**Table 2. Randomly sampled text feature metrics.**

| Sampled text feature metrics | Magazines texts | News texts |
|---|---|---|
| Number of sentences | 907,162 | 338,381 |
| Number of tokens | 18,223,083 | 5,961,326 |
| Average word length | 4.54 | 4.59 |
| Number of word types | 785,018 | 463,178 |
| Standard type-token ratio | 0.55 | 0.58 |
| Average sentence length | 20.09 | 17.62 |
| Average number of punctuations in one sentence | 3.37 | 3.56 |
| Average clause length | 5.96 | 4.95 |

**Table 3. Dependency distance per word description.**

| Word and its position code | Parent node word (governor) and position code | Dependency distance (absolute value) |
|---|---|---|
| The, 1 | Fox, 4 | 3 |
| Quick, 2 | Fox, 4 | 2 |
| Brown, 3 | Fox, 4 | 1 |
| Fox, 4 | Jumped, 5 | 1 |
| Jumped, 5 | ROOT | 0 |
| Over, 6 | Dog, 9 | 3 |
| The, 7 | Dog, 9 | 2 |
| Lazy, 8 | Dog, 9 | 1 |
| Dog, 9 | Jumped, 5 | 4 |

The calculation formula of NDD is:

$$NDD = abs\left(\ln\left(\frac{MDD}{\sqrt{rootDistance * sentenceLength}}\right)\right)$$

The root distance refers to the position code of the root word, and this index is ignored in MDD. The length of the sentence refers to the number corresponding to the largest position code of the sentence. This index is also ignored in MDD. Therefore, it can be said that the calculation of the NDD value is a calculation method that weights the sentence length and the root node position code based on MDD. In large-scale corpora, exceptionally long and short sentences are rare. The distribution of sentence length is a normal distribution around the average sentence length. For the most part, the left and right sides of the normal distribution can cancel each other out; accordingly, MDD values can still be relied upon with large data sets, which does not necessarily imply that NDD is better than MDD [22].

Aiming at the effectiveness of random sampling, this study conducted a full-text dependency syntactic analysis on part of the source documents in the COHA corpus and a correlation test. The corresponding numerical values calculated are significantly correlated, wherein the MDD Pearson correlation coefficient is 0.92 (>0.80, i.e., highly correlated), and the NDD Pearson correlation coefficient is 0.84 (>0.80, i.e., highly correlated), proving the validity and reliability of random sampling. Furthermore, random sampling did not significantly affect the results of the analysis in this study.

## Results and discussion

### Magazine texts

For magazine texts linear:

First, we observed the blue (MDD, Fig 2) and yellow (NDD, Fig 3) scatters as the diachronic distribution of the dependency distance of magazine texts. Both MDD and NDD of magazine texts show a decreasing trend. To better observe the diachronic linear trend of the dependency distance, the author performed linear regression calculations on the scatter plot of the diachronic distribution of the dependency distance.

The linear regression of both MDD and NDD of magazine texts shows a clear decreasing trend. Moreover, the author also examined the goodness of fit of MDD and NDD linear regression for magazine texts.

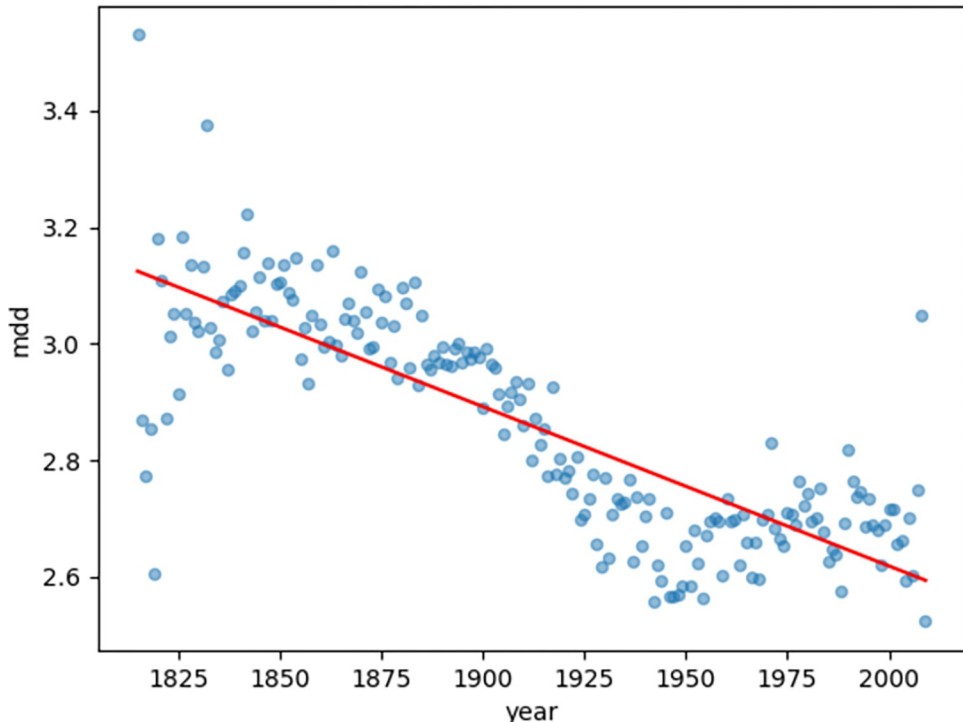

**Fig 2. Diachronic linear distribution of MDD for magazine texts.**

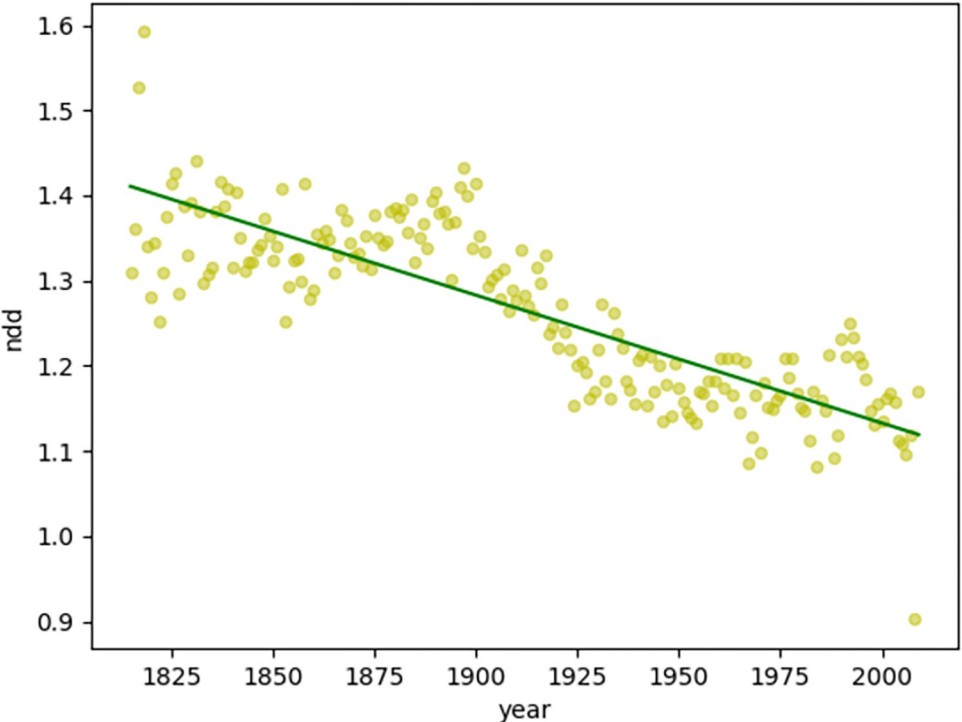

**Fig 3. Diachronic linear distribution of NDD for magazine texts.**

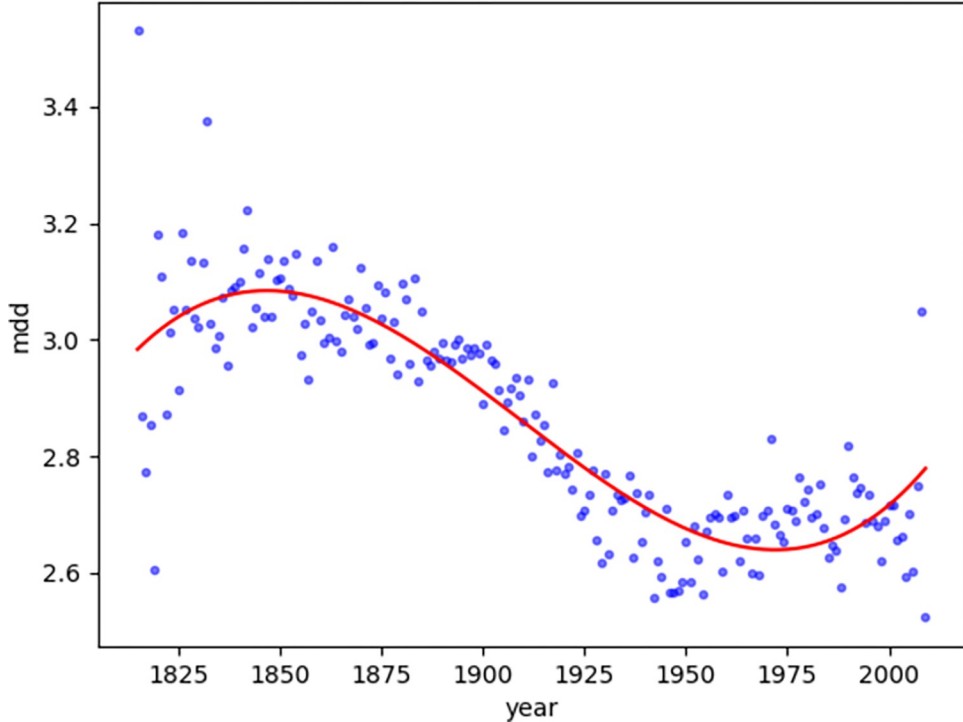

**Fig 4. Diachronic non-linear distribution of MDD for magazine texts.**

The goodness of fit refers to how well the regression line fits the observations. The statistic that measures the goodness of fit is called $R^2$ value. The maximum value of $R^2$ is 1. The closer the value of $R^2$ is to 1, the better the fit of the regression line to the observed distribution of scatters; in contrast, the lower the value of $R^2$, the worse the fit of the regression line to the observed distribution of scatters.

In Figs 2 and 3, the goodness of fit for MDD is 0.66 and the goodness of fit for NDD is 0.69. The author also calculated the goodness of fit of MDD and NDD in Lei's research [9], which were 0.68 and 0.69, respectively. The goodness of fit of MDD and NDD for magazine texts is high (>0.60), indicating a highly accountable decreasing trend.

For magazine texts non-linear:

In addition to a linear description that exhibits a clear trend, a non-linear polynomial regression [24] provides more fluctuation information. We can observe that the MDD (Fig 4) trend reaches 3.1 (with the highest value of 3.08 in year 1847) and starts to fall, indicating the validity of former research of 3 as a dependency distance limit [6]. Furthermore, the MDD trend reaches 2.6 (with the lowest value of 2.64 in year 1972) and starts to rise, indicating a certain bottom line of decrease, as opposed to 3 as the top limit.

In addition, the NDD (Fig 5) trend follows a similar trend as the MDD; the former reaches 1.4 (with the highest value of 1.37 in year 1844) and starts to fall, whereas the latter reaches 1.1 (with the lowest value of 1.14 in year 1999) and starts to rise.

## News texts

For news texts non-linear:

Second, we observed the diachronic distribution of the dependency distance of news texts. The time span of news texts is from 1860 to 2009, indicating that the genre of news developed

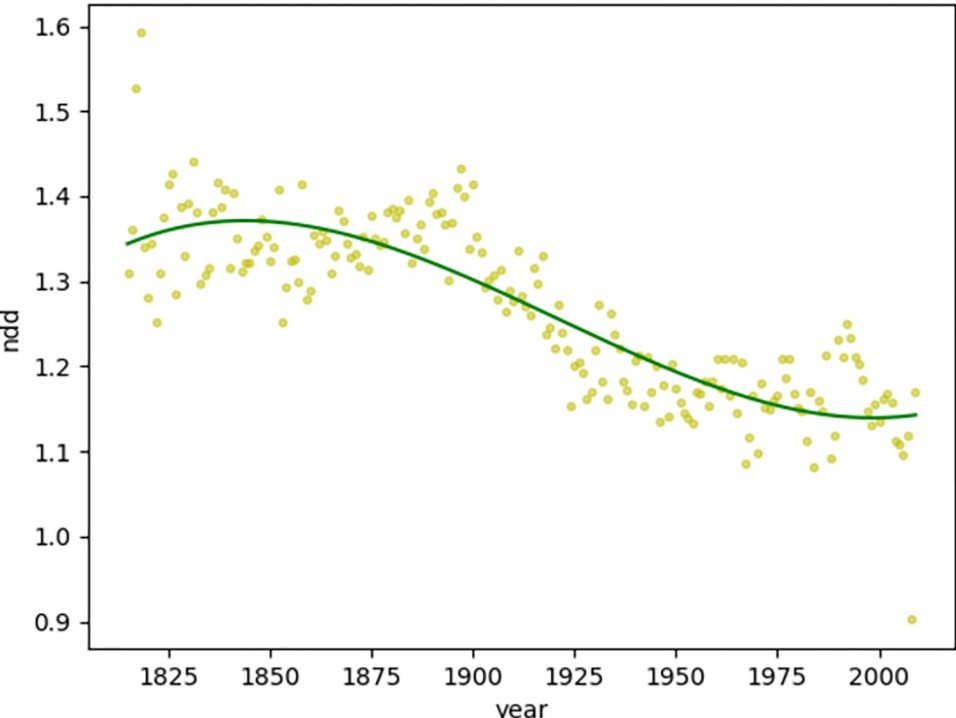

**Fig 5. Diachronic non-linear distribution of NDD for magazine texts.**

and matured from the mid-nineteenth century. News text has been popular and widespread ever since its appearance. News text serves not only as the dissemination of information but also as a representative genre for identifying and understanding information [25]. Because the linear regression goodness of fit of the diachronic distribution of the dependency distance of news texts is much lower (MDD: 0.14<0.60, NDD: 0.002<0.60), the author used nonlinear regression to examine the diachronic trends of news texts.

It can be observed that the overall diachronic trend of news texts is relatively stable. In particular, the NDD distribution is relatively uniform and there is no distinctive increase or decrease trend. The MDD (Fig 6) value fluctuates around 2.6 (with an average value of 2.61 and a median value of 2.59), and gradually rises at the end of the fluctuation, whereas the NDD (Fig 7) value fluctuates around 1.2 (with an average value of 1.20 and a median value of 1.20) (Table 4). The relatively young text genre of news has been widely disseminated with the appearance of newspapers. It has been committed to the cost-effectiveness of information transmission and has been using popular language to maximize information transfer conveniently.

When comparing the MDD and NDD of magazines texts and news text together, it can be noticed that the lowest point of the general downward trend of magazine texts coincides with the news texts, implying that the easy-to-understand news texts marks a certain boundary of the decreasing trend of dependence distance, or rather, the dependency distribution for magazines texts had been dropping closer to that of the news texts continuously in the past years.

With the assistance of the robust notion of information entropy [26], which can evaluate the information content as bits per character in written text [27], we adopted this algorism and carried it out on the sampled text of magazine and news text aforementioned. For two sentences with the same length of characters, the one with higher information entropy carries

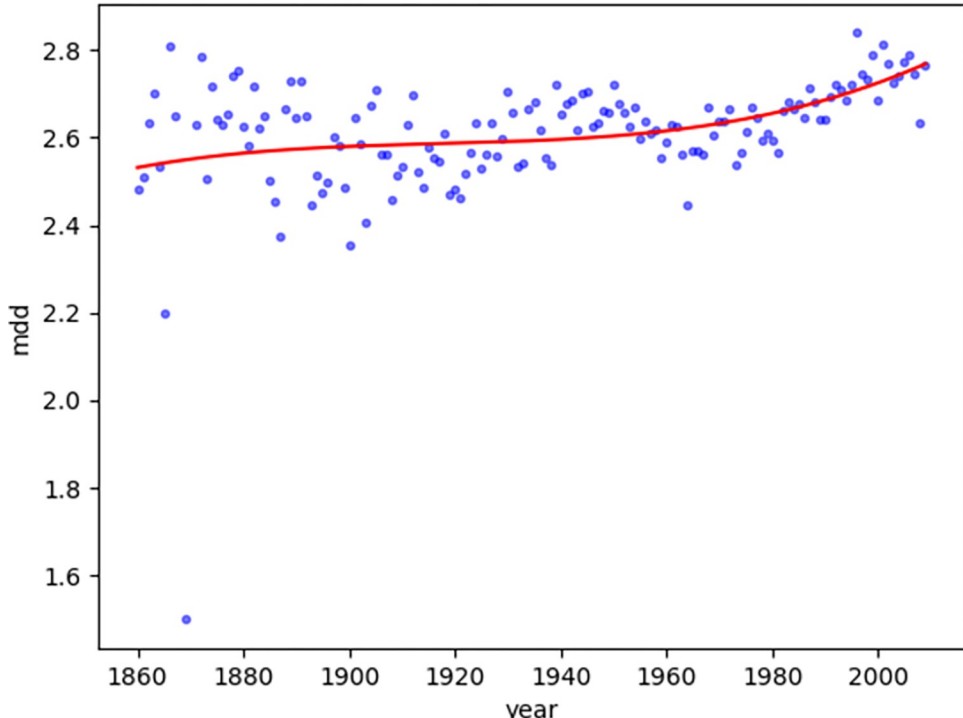

**Fig 6. Diachronic non-linear distribution of MDD for news texts.**

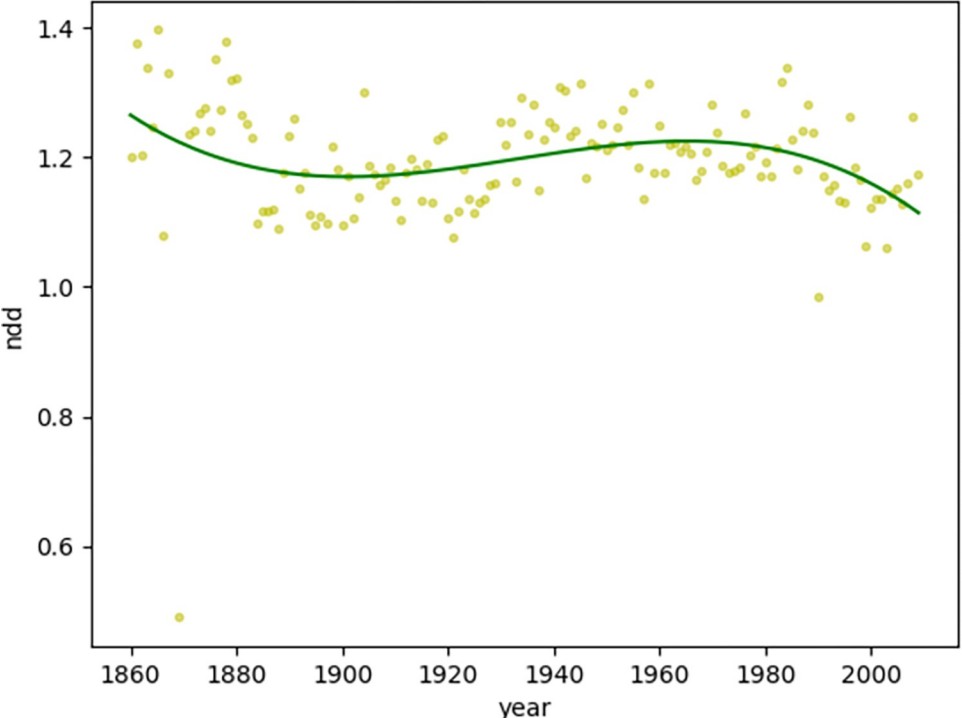

**Fig 7. Diachronic non-linear distribution of NDD for news texts.**

**Table 4. Observed MDD/NDD polynomial regression min/max/average/median values description.**

| Item | Min | Max | Mean | Median |
|---|---|---|---|---|
| MDD for Magazines | 2.64 | 3.08 | 2.86 | 2.85 |
| NDD for Magazines | 1.14 | 1.37 | 1.27 | 1.28 |
| MDD for News | 2.53 | 2.77 | 2.61 | 2.59 |
| NDD for News | 1.11 | 1.26 | 1.20 | 1.20 |

**Table 5. Information entropy comparison between magazines and news text.**

| Information Entropy | Value |
|---|---|
| Bits per Character for Magazines | 15.30 |
| Bits per Character for News | 14.69 |

more information content within, which indicates a higher efficiency of information transmitting.

Thus it becomes clear that the magazines text while maintaining higher language complexity with a mean value of MDD of 2.86 (NDD of 1.27), which is 1.09 times of the mean value of MDD of 2.61 (1.06 times of the mean value of NDD of 1.20) for news text, arrives at 15.30 bits per character of information entropy, only 1.04 times of the information entropy of 14.69 bits per character for news text (Table 5). The benefit of complexity increase does not bring as much benefit of transmitting more information, which concludes that the news text better balances language complexity and communication efficiency than magazine texts. This can also help to explain the observed trend in previous discussion that the trend of MDD and NDD of magazines texts are coming closer to that of news text in later years among the collected data. After all, the principle of labor-saving is to achieve the purpose of communication. The language of news is to a large extent the language that readers are most familiar with, and using familiar language in news helps to determine the credibility of the people and things on the newspaper page [28].

## Conclusion

Some studies have proved that political speeches show a significant diachronic dependency distance minimization trend. By considering the text as a whole with the parameters of MDD and NDD into account, this study discovered that the same is true for magazine text types but not for news text types, which remain relatively stable.

Compared with the synchronic investigation of dependence distance, which can define the upper limit of the dependence distance as 3 [6], the diachronic investigation of dependence distance provides inspiration to explore the central axis of its fluctuations from a diachronic perspective. The investigation of news texts and magazine texts in this study suggests that the MDD value fluctuates up and down with 2.6–2.9 as the central axis, with 2.5–3.1 as the boundary, whereas the NDD value fluctuates up and down with 1.2–1.3 as the central axis, with 1.1–1.4 as the boundary.

In future research, it is necessary to further expand and observe whether there is a diachronic decrease in dependency distance in more text types, larger datasets, and different languages, and whether there is a diachronic fluctuation axis of dependency distance, and moreover, what is the stable value of the central axis.

In addition, as magazines could be called elite media, whereas low-quality, highly shareable news could be regarded as mass media [29]. The research results have revealed that the language complexity of magazines, elite media, has now approached that of mass media.

## Supporting information

**S1 File.**
(ZIP)

## Author Contributions

**Writing – original draft:** Ruoyang Zhang.

**Writing – review & editing:** Guijun Zhou.

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
