## [Decision Letter · Decision Letter 0]

2 Oct 2022

PONE-D-22-24114An investigation of the diachronic trend of dependency distance in news and magazinesPLOS ONE

Dear Dr. ZHOU,

Thank you for submitting your manuscript to PLOS ONE. After careful consideration, we feel that it has merit but does not fully meet PLOS ONE’s publication criteria as it currently stands. Therefore, we invite you to submit a revised version of the manuscript that addresses the points raised during the review process.

We look forward to receiving your revised manuscript.

Kind regards,

Claudia Felser, Ph.D

Academic Editor

PLOS ONE

“Unfunded studies”

3. Thank you for stating the following in the Competing Interests/Financial Disclosure * (delete as necessary) section:

“NO authors have competing interests”

We note that you received funding from a commercial source: [Name of Company]

4. Please ensure that you include a title page within your main document. You should list all authors and all affiliations as per our author instructions and clearly indicate the corresponding author.

Additional Editor Comments :

Although the reviewers were both essentially positive about your manuscript, they have made several useful suggestions for improvement. Among other things they ask you to make the significance of your findings clearer, in particular in relation to the findings reported in Lei and Wen's (2020) study. They also ask for an improved discussion of the difference between the text types you examined, and of how news texts manage to balance complexity and efficiency. Reviewer 2 also makes a number of other suggestions that you should consider in your revisions.

I also have some additional comments of my own, which I am asking you to address in your revisions.

1. The abstract should be revised and re-focused. At present it focuses on Lei's work but says very little about the current study. The abstract also contains some unexplained abbreviations (MDD and NDD).

2. The introductory part contains several unclear wordings and attributions. E.g.

l.102 "One direction..." What do you mean here? Direction of current research?

l.110 "The study..." - What study are you referring to here?

l.117 "in the monolingual corpus and the hybrid corpus..." - What corpora are these? They have not been mentioned before. And what is meant by "hybrid"?

l.132 "young people's" - Is this finding specific to young people, and why do young people specifically matter for your study?

l.148 "The last aspect of the relevant study..." - What study are you referring to here?

3. Please make sure that all figures and tables are explicitly referred to in your main text.

4. Your title should be changed so as to include the term "dependency distance minimization" or "minimization of dependency distance". The current wording "diachronic trend of dependency distance" is uninformative.

Reviewers' comments:

Reviewer's Responses to Questions

**Comments to the Author**

1. Is the manuscript technically sound, and do the data support the conclusions?

Reviewer #1: Yes

Reviewer #2: Yes

2. Has the statistical analysis been performed appropriately and rigorously? 

Reviewer #1: Yes

Reviewer #2: Yes

3. Have the authors made all data underlying the findings in their manuscript fully available?

Reviewer #1: Yes

Reviewer #2: No

4. Is the manuscript presented in an intelligible fashion and written in standard English?

Reviewer #1: Yes

Reviewer #2: Yes

5. Review Comments to the Author

Reviewer #1: The study closely followed Lei and Wen (2020), and examined the dependency distance minimization in a diachronic corpus of news and magazine texts. It is well designed and clearly written. I would suggest publication after the following issues are addressed.

1. While the study is a close replication of Lei and Wen (2020) and much similar results were obtained, significance of the study should be offered.

2. The authors stated that "This research also indicated that news text could be a text type that best balances language complexity and communication efficiency." in the abstract. However, I did not see anything related to this important argument, particularly those pertinent to "communication efficiency" in the body of the article.

Reviewer #2: Review on

An investigation of the diachronic trend of dependency distance in news and magazines

This study quantitatively investigates the principle of dependency distance minimization (DDM) based on COCA. With special attention paid to the genres of news and magazines, it was found that both of these two text types witnessed some fluctuations diachronically in terms of the index of MDD and NDD, especially the text type of news. Moreover, the present study suggests that news texts achieve a balance between language complexity and communication efficiency.

Generally, the current work makes some exploration into the diachronic changes of the memory burden of these two text types. I appreciate that the authors make a good explanation of the theoretical background of the research and demonstrate their results clearly and concisely. However, one weak point of the work is the insufficient explanations of the reasons for the fluctuation of MDD and NDD with more references and citations to reach the conclusion that the text type of news cherishes a balance between complexity and efficiency.

Anyway, the article is a typical quantitative linguistic study that brings new findings to the diachronic changes of dependency distance and new insights into how MDD/NDD comes into force on different text types.

I recommend accepting this study but the comments, which I itemize below, might be taken into consideration.

Comments on some specific sections:

Title

It might be better to change the title from “An investigation of the diachronic trend of dependency distance in news and magazines” to “An investigation of the diachronic trend of dependency distance in magazines and news”, since in the Results and Discussion part, the analysis of magazines are presented first, and then magazines. Also, please check through the whole text the order of “magazines” and “news”, including Tables.

Abstract

Line 10: “inspection indicators of MDD and NDD” > please give the full name of MDD and NDD in the abstract since they are not so familiar to the general readers of PLOSone.

Section 1 Introduction

Line 102: “language, compared with the dependency distance of languages randomly generated” > two spaces between “languages” and “randomly” should be deleted.

Line 101-159: The author grouped quantitative dependency studies into 7 different types. However, it is not so clear in terms of the differences among so many groups. I recommended dividing this part into different themes. For example, it can be divided into the factors that may affect dependency distance (e.g., Jiang and Liu 2015; Wang and Liu 2017; Yan and Liu 2021), the mechanism of dependency distance minimization and its impact (e.g., Liu 2007; Lu, Xu and Liu 2016), and the application of dependency measures and quantitative features of dependency structures (e.g., Liu, Zhao & Li 2009; Liu and Xu 2012; Jing and Liu 2015). Then, it comes to the lead-in part of the topic of the current research.

Section 2 Research Methods

Line 182-183: change the font of Table 1 to be consistent with the whole text or the format requirement of PLOSone.

Line 213: add a citation to “Standard CoreNLP 4.4.0”, e.g., Manning et al. 2014 (maybe).

Line 221-223: “Furthermore, considering the huge volume of the COHA corpus, the author randomly selected 30% of the original corpus for analysis and generated an annotated corpus for data analysis based on the randomly sampled text.” > Add information on the final raw data adopted after the data pre-processing and random selection. A table similar to Table 1 might be favorable.

Line 224-225: “the average dependency distance MDD and the normalized average dependency distance NDD.” > (1) The expression should be reformulated. e.g., “the average dependency distance (MDD) and the normalized average dependency distance (NDD)”; (2) the citation should be added after MDD (Liu 2008) and NDD (Lei and Jockers 2020).

Line 228: punctuation mistake (please check through the whole text): the lazy dog,” > dog.”,

Line 232: In Table 2, row 1, column 2, “Parent node word(governor)” > “Parent node word (governor)”

Line 242: “normalized NDD” > the “N” means normalized.

Line 247: “MDD values can still be relied upon with large data sets”> two spaces between “values” and “can” need to be deleted.

Line 251: “The corresponding numerical values calculated are significantly” > two spaces between “values” and “calculated” need to be deleted.

Line 252: “Pearson correlation coefficient” > please check the assumptions of Pearson correlation. If the assumptions are not met, please use Spearman correlation.

Line 252: 0.922 > please explicit the range of high, medium and low coefficients, respectively.

Section 3 Results and Discussion

Line 267: the present author > the author or the present study

Line 274: R² > italicize statistics (R²)

Line 280-281: “The goodness of fit of MDD and NDD for magazine texts is high” > what is “high”? Give the value ranges for good, medium and low coefficients of R square.

Line 320-335: The discussion is interesting and provoke-thinking. However, one very important shortcoming of this part is that more information on the features of news and magazines, especially more explanations of the characteristics of the news text type to reach a balance between syntactic complexity and communication efficiency, is highly needed.

Section 4 Conclusion

Line 341-342: “Not all text types can reflect the diachronic dependency distance minimization trend” > this sentence is a little bit strange. You may delete it or add more explanations to this sentence.

Line 347: “ the upper limit of the dependence distance as 3” > should it be 4, rather than 3? Please check it and add a citation.

Line 359: “could be regarded as mass media. [27]” > wrong placement of the punctuation mark, it should be “could be regarded as mass media [27] .”

Line 359-361: “The research results have revealed that the language complexity of magazines, elite media, has now approached that of mass media.” > add more explanation of this sentence in Results and Discussion so as to reach this conclusion. It is very important.

In all, the current study focuses on the diachronic changes of MDD and NDD based on news and magazine texts from the COCA database. It shows that, rather than demonstrating a general trend of dependency distance minimization, these two text types witness great fluctuations over time. Besides that, the authors furthered their discussion in terms of the synergetic relations between syntactic complexity and communication efficiency. The authors not only show the quantitative results of the analyses, but also verifies the findings with statistical tests. However, there are still some points that need to be modified as I mentioned above. I sincerely hope these comments will help during the revision process.

6. PLOS authors have the option to publish the peer review history of their article (what does this mean?). If published, this will include your full peer review and any attached files.

Reviewer #1: No

Reviewer #2: No

---

## [Author Response · Author response to Decision Letter 0]

8 Dec 2022

Dear editor Claudia Felser,

On behalf of my co-authors, we thank you very much for giving us an opportunity to revise our manuscript. We appreciate editor and reviewers very much for their positive and constructive comments and suggestions on our manuscript entitled “An investigation of the diachronic trend of dependency distance in news and magazines”（PONE-D-22-24114）.

We are very sorry to update the revised manuscript so late because the adding of some necessary experiment data. Especially we have to research related concept of information enthropy which is very new to us. 

In this revised version, we have addressed the concerns of the reviewers. An item-by-item response to the reviewers' comments is enclosed, and the revision was marked in red fronts in the manuscript. We hope that these revisions successfully address their concerns and requirements and that this manuscript will be accepted. 

Looking forward to hearing from you soon.

Warm Regards

Guijun ZHOU

PhD, Professor of Linguistics, Northeast Normal University, PRC

---

## [Editor Report · Decision Letter 1]

15 Dec 2022

An investigation of the diachronic trend of dependency distance minimization in magazines and news

PONE-D-22-24114R1

Dear Dr. ZHOU,

We’re pleased to inform you that your manuscript has been judged scientifically suitable for publication and will be formally accepted for publication once it meets all outstanding technical requirements.

Kind regards,

Claudia Felser, Ph.D

Academic Editor

PLOS ONE
---

## [Editor Report · Acceptance letter]

20 Dec 2022

PONE-D-22-24114R1 

An investigation of the diachronic trend of dependency distance minimization in magazines and news 

Dear Dr. Zhou:

I'm pleased to inform you that your manuscript has been deemed suitable for publication in PLOS ONE. Congratulations! Your manuscript is now with our production department. 

Kind regards, 

on behalf of

Dr. Claudia Felser 

Academic Editor

PLOS ONE